# Effects of Heavy Metal Exposure on Shipyard Welders: A Cautionary Note for 8-Hydroxy-2′-Deoxyguanosine

**DOI:** 10.3390/ijerph16234813

**Published:** 2019-11-29

**Authors:** Ting-Yao Su, Chih-Hong Pan, Yuan-Ting Hsu, Ching-Huang Lai

**Affiliations:** 1Graduate Institute of Life Sciences, National Defense Medical Center, Taipei 114, Taiwan; timothy80329@mail.ndmctsgh.edu.tw (T.-Y.S.); misara116@mail.ndmctsgh.edu.tw (Y.-T.H.); 2Institute of Labor, Occupational Safety and Health, Ministry of Labor, New Taipei City 221, Taiwan; chpan@mail.ilosh.gov.tw; 3School of Public Health, National Defense Medical Center, Taipei 114, Taiwan; 4National Institute of Environmental Health Sciences, National Health Research Institutes, Miaoli 350, Taiwan

**Keywords:** shipyard, welding fumes, urinary heavy metals, urinary 8-hydroxy-2’-deoxyguanosine

## Abstract

Oxidative stress plays a crucial role in the development of diseases induced by welding fumes. To our knowledge, little information is available on the relationship between multiple heavy metal exposure and oxidative stress in welders. We assessed the relationship between multiple heavy metal exposure and oxidative damage by analyzing 174 nonsmoking male welders in a shipyard. Urinary metals were used as the internal dose of exposure to metals in welding fumes, and urinary 8-hydroxy-2’-deoxyguanosine (8-OHdG) was used as an oxidative DNA damage marker. The relationship between workers’ metal levels and 8-OHdG was estimated using a multiple linear regression model. The geometric mean levels of urinary chromium (Cr), nickel (Ni), cadmium (Cd), and lead (Pb) were considerably higher in welders than in controls. Urinary Cr and Ni were determined as effective predictors of urinary 8-OHdG levels after adjusting for covariates. Oxidative DNA damage was associated with both Cr and Ni of welding fume exposure in shipyard welders (Ln Cr: β = 0.33, 95%C.I. = 0.16–0.49; Ln Ni: β = 0.27, 95%C.I. = 0.12–0.43). In this study, we investigated the significantly positive relationship between urinary metals (especially Cr and Ni) and 8-OHdG in nonsmoking shipyard welders. Moreover, the use of particulate respirators did not reduce metal exposure and oxidative damage. Therefore, we infer that hazard identification for welders should be conducted.

## 1. Introduction

Welding fumes were classified as a possible human carcinogen (group 2B) by the International Agency for Research on Cancer (IARC) in 1990 [1], and an evaluation update for carcinogenicity was recommended by an IARC expert group in 2010 [2]. Welding fumes are recognized as a group 1 carcinogen in humans because numerous studies have reported an increased risk of lung cancer due to such fumes [3,4]. In addition, chronic welding fume exposure has been reported to exhibit a relationship with lung cancer in welders with no or mild smoking habits [5], as well as pharynx and larynx cancer [6,7,8].

Welding fumes consist of many toxic materials, including several metals such as chromium (Cr), nickel (Ni), cadmium (Cd), and lead (Pb) [9], which potentially and adversely affect welders’ health. Shipbuilding and repairing involve extensive welding processes [10]. Most welding fumes comprise oxidized metal particles of respirable fraction size that are produced during the joining of metal pieces by arc welding at an extremely high temperature from heated parent materials and fillers/fluxes [9,11,12,13].

The most common DNA lesion caused by the reaction of hydroxyl radicals with guanosine at the C-8 position in DNA is 8-Hydroxy-2’-deoxyguanosine (8-OHdG) [14]. DNA damage may be repaired using the base excision repair pathway, and the resulting repaired product, urinary 8-OHdG, is not affected by diet or cell turnover [15]. Heavy metals, such as Cr, Ni, Cd, and Pb, may produce reactive oxygen species (ROS), such as hydroxyl radicals (OH), superoxide anion (O_2_^−^), singlet oxygen (^1^O_2_), and H_2_O_2_ [16,17]. The increased production of ROS can induce several key signaling events, which can cause adverse effects of oxidative stress and lead to diseases [18]. Such damage is associated with cancer [19,20].

For the last several decades, epidemiologists have been warning of the hazards of welding fumes; however, until now, welding fume-induced health problems remain an occupational hazard that must be solved. The adverse effects of heavy metals from welding fume exposure are observed in both humans and animals [21,22,23]; however, a comprehensive understanding of oxidative stress occurring in multiple metal exposure populations remains unclear. Moreover, limited information on the use of personal respiratory protection equipment (RPE) is available. Therefore, this study focused on the identification of oxidative stress-based biomarkers associated with exposure to heavy metals in welding fumes.

## 2. Materials and Methods

### 2.1. Study Population

We conducted a cross-sectional study to assess the relationship between metal exposure and oxidative damage. The study participants were recruited from a shipyard located in Taiwan during their annual physical examination. We collected their spot urine for urinary metal, 8-OHdG, and creatinine analyses. Sociodemographic information, such as sex, age, education, and lifestyle (e.g., smoking habits, alcohol intake, and betel nut chewing), was obtained using a self-reported questionnaire. The history of diagnosed illness, information on chemical exposure, and working experience with a specific welding fume exposure, including the frequency of the particulate respirator use, were also obtained from the questionnaire. Participants with highly diluted (urinary creatinine < 30 mg/dL) or highly concentrated (urinary creatinine > 300 mg/dL) urine samples were also excluded from our statistical analysis according to the WHO guidelines for the biological monitoring of chemical exposure in the workplace [24,25]. Moreover, since cigarette smoking is a strong confounding factor of welding fume exposure and 8-OHdG is potentially confounded by diagnosed cancer, participants with a cigarette smoking habit and diagnosed cancer were excluded. Finally, 174 nonsmoking shipyard welders were selected for the analysis and were classified into two groups, namely welders and office workers based on the job titles obtained from responses to the questionnaire. The study protocol was approved by the Institutional Review Board of Tri-Service General Hospital. All participants provided oral and written informed consent for providing information for the study.

### 2.2. Assessment of Exposure to Metals in Workplace Air

Personal air samples from the workplace were collected using active samplers with cellulose-ester filters (diameter: 37 mm and pore size: 0.8 μm). Metals were sampled at a flow rate of 2000 ± 30 mL/min. Personal samplers were placed nearly at a height of the breathing level of workers in welding areas for consecutive 8-hour workdays. The concentrations of Cr, Ni, Cd, and Pb were measured through inductively coupled plasma mass spectrometry (ICP-MS, Agilent 7500ce, Agilent, Santa Clara, CA, USA). The detection limits for Cr, Ni, Cd, and Pb were 5.2 ng/L, 6.3 ng/L, 1.2 ng/L, and 8.0 ng/L that were obtained using seven repeated analyses of deionized water.

### 2.3. Urinary Biomarker Analysis

Participants were asked to wash their hands with soap prior to urine collection; spot urine samples were collected after the shift ended on a weekend. Urine samples were stored at −20 °C until analysis. Urinary metals, namely Cr, Ni, Cd, and Pb, were analyzed through ICP-MS (Agilent 7500ce, Agilent, Santa Clara, CA, US). Urinary 8-OHdG levels were measured through high-performance liquid chromatography-tandem mass spectrometry (HPLC/MS/MS) as described in [26]. A detection limit of 5.7 ng/L was obtained using seven repeated analyses of deionized water. The intraday and interday coefficients of variation were 2–3% and 4–5%, respectively [27]. The urinary creatinine level was determined using an automated method based on the Jaffe reaction for urinary biomarker correction [25,28].

### 2.4. Statistical Analysis

Urinary 8-OHdG, metals, and creatinine levels were first natural log-transformed to normalize their distributions before the Student’s *t*-test or regression analysis was done. Student’s t and Chi-Square statistics were used to compare the personal covariates, which were the urinary 8-OHdG and metals of welders and office workers. The association among urinary metals, personal covariates, work experience, and oxidative damage biomarkers was assessed using multiple linear regression; moreover, to minimize the effect of residual confounding of sociodemographic factors, such as sex, age, and race/ethnicity, on urinary biomarkers, urinary creatinine was adjusted as an independent variable in the linear regression analysis [25]. The level for statistical significance was set to α = 0.05 in all tests. Statistical analysis was conducted using IBM SPSS statistics software for Windows version 22.0 (IBM Corp., Armonk, NY, USA).

## 3. Results

The characteristics of the study participants are presented in Table 1. A total of 121 male welders and 53 male office workers were included in the study. The age, seniority, height, weight, body mass index, and urinary creatinine level of welders were similar to those of office workers (*p* > 0.05). The proportion of education above college was higher in office workers (84.91%) than in welders (26.45%) (*p* < 0.001), and approximately 44% of welders regularly wore particulate respirators during their work shift.

The metal concentrations in ambient air were obtained using personal air sampling as listed in Table 2. The geometric mean and geometric standard deviation of ambient Cr, Ni, Cd, and Pb were 5.27 and 3.28 μg/m^3^; 6.47 and 5.29 μg/m^3^; 0.40 and 2.95 μg/m^3^; and 5.39 and 4.24 μg/m^3^, respectively.

Table 3 presents a comparison of urinary biomarkers between welders and office workers. The geometric mean concentrations of urinary Cr, Ni, Cd, Pb, and 8-OHdG in welders were 2.06, 3.13, 0.69, 4.18, and 4.77 μg/g creatinine, respectively, and those in office workers were 0.74, 1.66, 0.56, 2.86, and 2.54 μg/g creatinine, respectively. After conducting measurements using natural log transformation and Student’s *t*-test, a considerably higher amount of urinary biomarkers was observed in welders than in office workers.

The relationship between the levels of heavy metals in air and urine samples was determined using the Pearson correlation analysis (Table 4). Individual air Cr, Ni, and Cd levels were positively related to individual urinary Cr, Ni, and Cd levels, Pearson correlation coefficients were 0.34, 0.33, and 0.21, respectively. However, the individual air Pb level was not significantly correlated to the individual urinary Pb level.

Figure 1 presents a comparison of urinary biomarkers by welders/office workers and particulate respirator usage. The concentration of all urinary biomarkers was considerably higher in welders than in office workers. By contrast, the concentration of urinary biomarkers did not differ in welders working with or without regular particulate respirators.

Table 5 lists the Pearson correlation matrix across urinary biomarkers including metals, 8-OHdG, and urinary creatinine. The correlation coefficients were statistically positive and significant among metals, such as Cr to Ni (r = 0.50), Cr to Cd (r = 0.31), Cr to Pb (r = 0.31), Ni to Cd (r = 0.52), Ni to Pb (r = 0.26), and Cd to Pb (r = 0.43). In addition, urinary metals including Cr, Ni, and Cd were investigated for a significantly positive correlation with urinary 8-OHdG (r = 0.42, 0.41, and 0.18, respectively).

Table 6 demonstrates the results of the multiple linear regression obtained after adjusting for potential confounders, namely age, BMI, and urinary creatinine, which facilitate the determination of urinary heavy metals and 8-OHdG. Compared with office workers, the geometric mean of urinary Cr, Ni, and 8-OHdG were 148.43%, 71.60%, and 68.47% higher in welders who did not regularly use particulate respirators, respectively, whereas the geometric mean of urinary Cr, Ni, Pb, and 8-OHdG were 153.45%, 76.83%, 40.49%, and 78.60% higher in welders who regularly used particulate respirators, respectively. Notably, urinary metal and 8-OHdG concentrations were slightly higher in welders who regularly used particulate respirators than in those who did not. Additionally, both single and multiple metal models were analyzed to evaluate the contribution of urinary metals to urinary 8-OHdG. In single metal models, urinary 8-OHdG was increased by 37.55%, 30.13%, and 17.28% per doubling urinary of Cr, Ni, and Cd, respectively. Moreover, in multiple metal models, urinary 8-OHdG was increased by 25.70% and 20.58% per doubling of urinary Cr and Ni, respectively. The decrease in 8-OHdG was non-significantly related to urinary Cd and Pb.

## 4. Discussion

Urinary Cr, Ni, and Cd levels were positively correlated with the 8-OHdG concentration. The geometric mean 8-OHdG concentration (5.02 μg/g creatinine) in welders in this study was higher than that in gas metal arc welders working with stainless steel, flux-cored arc welders working with mild steel, tungsten inert gas welders, and welders wearing a powered air-purifying respirator in an earlier study (median = 3.79–4.41 μg/g creatinine) [29], as well as welders in another study (median = 3.97 μg/L) [30]. Compared with other occupational metal exposure, the median urinary 8-OHdG concentration was considerably higher in workers exposed to hexavalent Cr electroplating (median = 13.65 μg/g creatinine) than in welders in our study [31]; in another study involving workers exposed to hexavalent Cr electroplating, the urinary 8-OHdG concentration was higher than that in our study (mean = 20.73 μg/L) [32]. Our study supports the finding that urinary 8-OHdG levels can be reliably used for assessing exposure to Cr and Ni in shipyard welders. A recent study reported that the cancer risk of occupational Cr exposure exceeded the acceptable 10^−3^ level for occupational exposure in shipbuilding welders, rather than Ni [33].

Welding fumes contained various contaminants generated from heated metal work pieces and flux materials. Some of the metals and fluxes were vaporized and then condensed as nanometer-sized particles [34,35,36,37]. In humans, inhaled ultrafine particles (aerodynamic diameters < 0.1 μm) deposit in the lower respiratory tract and penetrate the alveolar–blood barrier rapidly entering the circulatory system. [38] Moreover, welding fumes contain many toxic metals, such as Cr, Ni, Cd, and Pb. Vaporized metals were oxidized in air and then inhaled. Various welding fume components exhibit different toxicological properties. An epidemiological study demonstrated that exposure to PM_2.5_ during the welding process was related to increased urinary 8-OHdG levels [21].

Oxidative stress occurs through the imbalance of free radicals and antioxidants due to either the depletion of antioxidants or the accumulation of free radicals. Consequently, free radicals attack and oxidize cell components, such as lipids, proteins, and nucleic acids, resulting in cell dysfunction [18,39,40]. Metals such as Cr induce oxidative stress through the Haber–Weiss/Fenton reaction and produce superoxide radicals, hydroxyl radicals, and other ROS. In addition, metals such as Ni, Cd, and Pb induce oxidative stress by stimulating the activity of NADPH oxidase or depleting glutathione and bonding to the sulfhydryl group of proteins [39,40,41,42]. Consequently, the increased generation of ROS damages DNA [17]. The most common product of DNA lesion is 8-OHdG, which is produced during the repair of oxidatively damaged DNA in vivo [43,44]. Epidemiological studies have reported that urinary 8-OHdG is associated with cancer [19,20].

The half-life of urinary Cr excretion was 10.75 years in workers who performed the plasma cutting of stainless steel [45]. After ceasing environmental exposure, the half-life of urinary Cd was 16 years in men [46]. By contrast, the half-life of urinary Cd was 12 years in welders working with stainless steel [47]. Therefore, urinary Cr and Cd are long-term biomarkers for the detection of welding fume exposure extent. By contrast, urinary Ni exhibited a short half-life of 96 hours in welders exposed to stainless steel welding fumes [48]. A controlled human exposure study indicated that after one hour of controlled tungsten inert gas welding fume exposure, a considerable increase in urinary 8-OHdG was observed in workers three hours after their shift ended [22].

The results of the WELDOX study group indicated increased urinary Cr and Ni concentrations in welders working in confined spaces and decreased urinary Cr and Ni concentrations in welders working in sufficient ventilation and using respiratory mask [49]. Additionally, after introducing interventions, such as welding helmet use and a regularly purified air supply, for improving the ventilation and respiratory protection, a decrease in urinary Cr and Ni concentrations was observed during a follow-up after three years [50]. In this study, the use of particulate respirators was not an effective predictor of both urinary metals and 8-OHdG. This finding is not consistent with that of an earlier study, which indicated that using the particulate respirator considerably reduced the malondialdehyde (MDA) amount in exhaled breath condensate of welders [51]. A possible explanation for this is that the incorrect method of wearing particulate respirators may increase urinary metal and 8-OHdG concentration, despite welders regularly wearing particulate respirators due to dispersed hazardous particles in the workplace. To reduce welding fume exposure, methods for using RPE and the cartridge or canister should be renewed before breakthrough by following the guidelines. In this study, the effects of welding fumes were identified and RPE use among participants was improved.

We investigated the relationship between multiple metal exposure and urinary 8-OHdG; however, some limitations were observed. First, the study failed to measure some potentially useful elements, such as volatile organic carbon and other metals including iron and zinc [21]. Moreover, information on dietary exposure to Cr and Ni were not collected [52]. Another limitation was the difficulty in determining the causal-inference relationship by using a cross-sectional study design. Regardless of limitations, urinary Cr and Ni were found to be effective predictors of urinary 8-OHdG levels in male nonsmoking workers at the shipyard. Exposure to Cr and Ni increases the risk of oxidative DNA damage in shipyard welders.

## 5. Conclusions

In this study, the use of particulate respirators was not an effective predictor for both urinary metals and 8-OHdG in welders. The levels of urinary Cr, Ni, and 8-OHdG were significantly higher in welders than in office workers. The levels of Cr and Ni in urine samples were positively associated with the 8-OHdG levels of shipyard welders, indicating that preventive measures must be developed immediately, including the use of effective gloves and respirators and enforcement of ventilation to reduce welding fume exposure and promote the health of shipyard welders.

## Figures and Tables

**Figure 1 ijerph-16-04813-f001:**
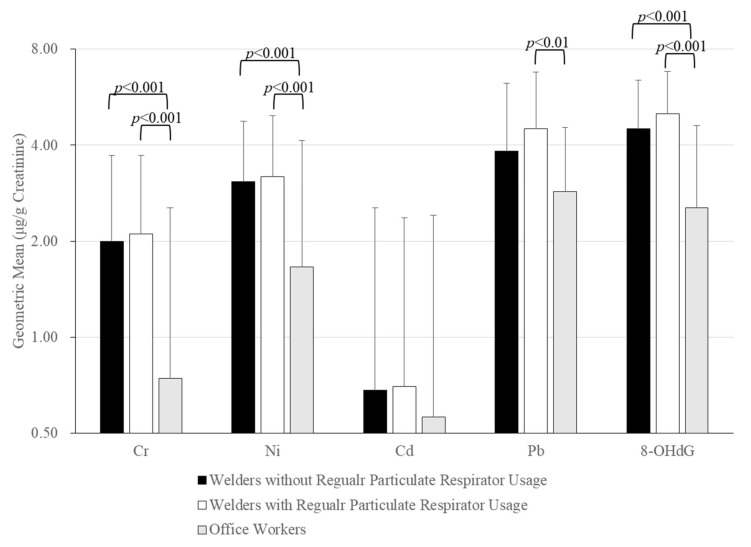
Comparison of urinary biomarkers between welders without/with regular particulate respirator use and office workers. The measurements were natural log-transformed and examined using Scheffe’s test after one-way analysis of variance (ANOVA) test.

**Table 1 ijerph-16-04813-t001:** Characteristics of study participants.

	Welders (n = 121)	Office Workers (n = 53)	*p*-Value
**Personal characteristics, mean ± SD**			
Age (years) ^a^	51.61 ± 7.67	51.25 ± 8.08	0.78
Seniority (years) ^a^	32.98 ± 9.76	30.08 ± 11.64	0.12
Height (cm) ^a^	168.43 ± 5.62	168.89 ± 7.23	0.69
Weight (kg) ^a^	68.34 ± 7.44	69.15 ± 9.41	0.58
Body Mass Index (kg/m^2^) ^a^	24.09 ± 2.36	24.21 ± 2.65	0.77
Urinary Creatinine (mg/dL, GM, GSD) ^c^	124.15, 1.68	137.01, 1.55	0.23
**Gender, n (%)**			
Male	121 (100.00)	53 (100.00)	
**Education, n (%) ^b^**			<0.001
Under High School	89 (73.55)	8 (15.09)	
Above College	32 (26.45)	45 (84.91)	
**Frequency of Particulate Respirator Usage, n (%) ^b^**			<0.001
Not Regularly	67 (55.37)	53 (100.00)	
Regularly	54 (44.63)	0 (0.00)	

^a^ Student’s *t*-test; ^b^ χ^2^ test; ^c^ Values were natural log-transformed and examined using Student’s *t*-test.

**Table 2 ijerph-16-04813-t002:** Concentrations of metals in personal air samples of welders (n = 106).

	GM, GSD	Median (Min–Max)
Cr (µg/m^3^)	5.27, 3.28	6.05 (0.15–116.70)
Ni (µg/m^3^)	6.47, 5.29	4.99 (0.05–68.67)
Cd (µg/m^3^)	0.40, 2.95	0.33 (0.02–32.85)
Pb (µg/m^3^)	5.39, 4.24	6.44 (0.31–56.78)

**Table 3 ijerph-16-04813-t003:** Comparison of urinary biomarkers between welders and office workers.

	Welders (n = 121)	Office Workers (n = 53)	*p*-Value ^a^
	GM, GSD	GM, GSD	
Urinary Cr (μg/g Creatinine)	2.06, 1.64	0.74, 1.80	<0.001
Urinary Ni (μg/g Creatinine)	3.13, 1.72	1.66, 2.48	<0.001
Urinary Cd (μg/g Creatinine)	0.69, 1.75	0.56, 1.85	<0.05
Urinary Pb (μg/g Creatinine)	4.18, 2.35	2.86, 1.68	<0.001
Urinary 8-Hydroxy-2′-Deoxyguanosine (μg/g Creatinine)	4.77, 1.84	2.54, 2.07	<0.001

^a^ Values were natural log-transformed and examined using Student’s *t*-test.

**Table 4 ijerph-16-04813-t004:** Pearson correlation coefficients for natural log-transformed heavy metals in personal air sampling total dust and urine of welders (n = 106).

	Urine	Ln Cr (µg/L)	Ln Ni (µg/L)	Ln Cd (µg/L)	Ln Pb (µg/L)
Total Dusts					
Ln Cr (µg/m^3^)		0.34 ***			
Ln Ni (µg/m^3^)			0.33 ***		
Ln Cd (µg/m^3^)				0.21 *	
Ln Pb (µg/m^3^)					0.15

* *p* < 0.05; ** *p* < 0.01; *** *p* < 0.001

**Table 5 ijerph-16-04813-t005:** Pearson correlation coefficients for natural log-transformed urinary biomarkers (n = 174).

	Ln U-Ni (μg/L)	Ln U-Cd (μg/L)	Ln U-Pb (μg/L)	Ln U-8-OHdG (μg/L)	Ln U-Creatinine (mg/dL)
Ln U-Cr (μg/L)	0.50 ***	0.31 ***	0.31 ***	0.42 ***	−0.02
Ln U-Ni (μg/L)		0.52 ***	0.26 ***	0.41 ***	0.07
Ln U-Cd (μg/L)			0.43 ***	0.18 *	0.19 *
Ln U-Pb (μg/L)				0.10	0.02
Ln U-8-OHdG (μg/L)					0.04

* *p* < 0.05; ** *p* < 0.01; *** *p* < 0.001

**Table 6 ijerph-16-04813-t006:** Determinants of urinary heavy metals and 8-hydroxy-2′-deoxyguanosine (n = 174).

	β (Lower–Upper)	GM % Change	*p*-Value
**Ln Urinary Cr (μg/L) ^a,b^**			
Welders without Regular Particulate Respirator Usage	0.91 (0.83–1.00)	148.43	<0.001
Welders with Regular Particulate Respirator Usage	0.93 (0.84–1.01)	153.45	<0.001
Office Workers	Reference		
**Ln Urinary Ni (μg/L) ^a,b^**			
Welders without Regular Particulate Respirator Usage	0.54 (0.35–0.73)	71.60	<0.001
Welders with Regular Particulate Respirator Usage	0.57 (0.39–0.76)	76.83	<0.001
Office Workers	Reference		
**Ln Urinary Cd (μg/L) ^a,b^**			
Welders without Regular Particulate Respirator Usage	0.12 (−0.03–0.27)	12.75	0.12
Welders with Regular Particulate Respirator Usage	0.13 (−0.01–0.27)	13.88	0.07
Office Workers	Reference		
**Ln Urinary Pb (μg/L) ^a,b^**			
Welders without Regular Particulate Respirator Usage	0.22 (−0.01–0.44)	24.61	0.06
Welders with Regular Particulate Respirator Usage	0.34 (0.13–0.56)	40.49	<0.01
Office Workers	Reference		
**Ln Urinary 8-Hydroxy-2′-Deoxyguanisine (μg/L) ^a,b^**			
Welders without Regular Particulate Respirator Usage	0.50 (0.33–0.68)	64.87	<0.001
Welders with Regular Particulate Respirator Usage	0.58 (0.41–0.74)	78.60	<0.001
Office Workers	Reference		
**Ln Urinary 8-Hydroxy-2′-Deoxyguanisine (μg/L) ^a,c^**			
Ln Urinary Cr (μg/L)	0.46 (0.31–0.61)	37.55	<0.001
**Ln Urinary 8-Hydroxy-2′-Deoxyguanisine (μg/L) ^a,c^**			
Ln Urinary Ni (μg/L)	0.38 (0.25–0.51)	30.13	<0.001
**Ln Urinary 8-Hydroxy-2′-Deoxyguanisine (μg/L) ^a,c^**			
Ln Urinary Cd (μg/L)	0.23 (0.04–0.43)	17.28	<0.05
**Ln Urinary 8-Hydroxy-2′-Deoxyguanisine (μg/L) ^a,c^**			
Ln Urinary Pb (μg/L)	0.08 (−0.05–0.21)	5.70	0.23
**Ln Urinary 8-Hydroxy-2′-Deoxyguanisine (μg/L) ^a,c^**			
Ln Urinary Cr (μg/L)	0.33 (0.16–0.49)	25.70	<0.001
Ln Urinary Ni (μg/L)	0.27 (0.12–0.43)	20.58	<0.001
Ln Urinary Cd (μg/L)	−0.06 (−0.27–0.15)	−4.07	0.58
Ln Urinary Pb (μg/L)	−0.05 (−0.17–0.08)	−3.41	0.46

^a^ Age (years), body mass index (kg/m^2^), and Ln urinary creatinine (mg/dL) were adjusted as covariates; ^b^ Geometric mean percent change (GM% Change) = (e^β^ − 1) × 100%; ^c^ Geometric mean percent change (GM% Change) = (2 ^β^ − 1) × 100%

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
