# Peer review of "Effects of Heavy Metal Exposure on Shipyard Welders: A Cautionary Note for 8-Hydroxy-2′-Deoxyguanosine"

_ijerph, 2019, doi:10.3390/ijerph16234813_

Round 1

Reviewer 1 Report

Dear authors,

After evaluating your manuscript, I strongly believe that you need to strengthen the results section for it to be publishable.  First, the results narrative is too short in a way that it does not clearly describe what is presented in the tables and the purpose of each particular analysis. 

Table 3: There is too much information in table 3, which include 3 different analyses, which it is not easy to understand as presented which comparisons are you making.  Maybe it will be better to separate the results into two sections or tables, one comparing office workers vs welders and the other for respirator use.  Otherwise it can be presented as graphs, to make it more clear.

Table 4:  The table was mentioned in the text but not included in the manuscript.  In addition, the narrative section of the results does not mention which correlations were observed and which were significant. And if the urinary metal concentrations were correlated with the air metal concentrations.

Table 5: You need to explain the regression model and the meaning of the beta coefficients.  Also you might need to do the regression analysis with the metals air concentrations.

For the references, which are used in the introduction and discussion please update with more recent ones.

Author Response

Dear Editor:

We would like to submit the revised manuscript titled “Effects of Heavy Metal Exposure on Shipyard Welders: A Cautionary Note of 8-Hydroxy-2’-Deoxyguanosine” to be published in the International Journal of Environmental Research and Public Health.

All comments have been considered and addressed carefully. All changes have been made using the "Track Changes" function in Microsoft Word in the revised manuscript.

On behalf of the research group,

Yours sincerely,

Ching-Huang Lai, Ph.D.

Professor

School of Public Health,

National Defense Medical Center,

161 Minquan East Road Section 6, Neihu,

114 Taipei City,

Taiwan.

Reviewer 2 Report

This is a very important piece of work. This information is necessary for the welding community to be able to make the changes required in order to reduce their heavy metal exposures.  So well-done on the study. 

However, the grammar and style of the paper needs to be improved significantly. It was very difficult to understand what you were trying to say due to grammatical errors. Please get it checked by a native English speaking person.

I am providing you with some overall comments and in some specific places as examples of the errors, but please rewrite this paper with the help of an English speaking person. 

Abstract: Grammatical errors. Please re-write this section with correct tenses and grammar. For example:

Line 11 - Change the words "is considered" to has. 

Line 13 - Change to "we assessed" 

Introduction: Please re-write this sections with correct grammatical terms and sentences. There are many grammatical errors which makes it difficult to understand and interpret. For example:

Line 29 - That sentence needs to be rephrased

Line 31 - The sentence is grammatically incorrect. It could be re-written as: Numerous studies have reported  an increased risk of lung cancer due to welding fumes as it is recognised as a group 1 carcinogen in humans. 

Line 33 - Re-write the sentence. 

Line 52 - Re-phrase please.

Materials and methods: Major grammatical errors. Some places require more information and clarifications. Please correct the sections. Its very hard to understand what you are trying to relay. For example: 

Line 62 - I am unsure what physical examination is. Please explain what physical measures you are  collecting.

Line 66 - Change  by the to "via"

Line 67 - Rewrite that sentence. I am not sure what you are trying to say. Are you saying that a separate analysis was considered for non smoking welders? 

Line 75 - Not sure what this sentence says. 

Line 86 - Re write this sentence

Line 94 - Reference this sentence

Results: Again requires grammatical check. Please re-write this section as it is a very important section. 

Line 105 - You say that welders and office workers were included in this research. This is not clarified in the methods section. 

Line 105 - change were to "are"

Line 106 - Job years: what job are you referring to? is it welders? Perhaps change this to "occupation" maybe? 

Line 111 and line 116 - I just noticed in some places you have added metal symbols and other places you spell them in full. Please be consistent. 

Line 119 - Rephrase 

Line 122 - 129  - Rephrase 

Tables - In some you have the metals measures as ug/m3 and in another ug/g. Please check. 

Discussion: Please re-write this section due to grammatical errors and poor sentence constructions. 

Conclusion: Please re-write this section. 

Author Response

(The authors gave the same response as above.)

Round 2

Reviewer 2 Report

In the abstract, the metals should be spelt out first followed by the symbol. E.g Chromium (Cr) etc. This can then be followed by symbols within the rest of the paper. I see that you have had the grammer checked which is great. Please re-check that the paper reads well and there are no further grammatical mistakes. 

Author Response

Manuscript ID: ijerph-628192

A Point-by-Point Response to Reviewer 2’s Comments to Author:

The research design can be improved.

Response: We agree with your comment. We have added the text “Since cigarette smoking is a strong confounding factor of welding fume exposure, only the 174 non-smokers were used as study subjects.“ to the “Study Population” at lines 59 to 61 on page 2 in the Materials and Methods section.

The methods can be improved.

Response: We agree with your comment. We have added the text “The level for statistical significance was set to α=0.05 in all tests.”to the “Statistical Analysis” and revised the “Statistical Analysis” at lines 94 to 99 on page 3 in the Materials and Methods section. Furthermore, we have added the text “The detection limits for Cr, Ni, Cd and Pb were 5.2 ng/L, 6.3 ng/L, 1.2 ng/L and 8.0 ng/L that were obtained using seven repeated analyses of deionized water.” to the “Assessment of Exposure to Metals in Workplace Air at lines 83 to 84 on page 2 in the Materials and Methods section.

The results can be improved.

Response: We agree with your comment. We have revised the descriptions of Table 4. and Table 5. In the Results section (Lines 119 to 120).

In the abstract, the metals should be spelt out first followed by the symbol. E.g. Chromium (Cr) etc. This can then be followed by symbols within the rest of the paper. I see that you have had the grammar checked which is great. Please re-check that the paper reads well and there are no further grammatical mistakes.

Response: Thank you for this comment. We have revised the expression of metals and its abbreviations in the Abstract section (Lines 16 to 17 on page 1). Moreover, thank you for your kind suggestions about the grammar correction previously, it is really helped. In addition, we have revised the manuscript and confirmed that there were no grammatical mistakes.

This manuscript is a resubmission of an earlier submission. The following is a list of the peer review reports and author responses from that submission.